# 1D versus 2D cocrystals growth via microspacing in-air sublimation

Xin Ye[1], Yang Liu[1], Qing Guo[1], Quanxiang Han[1], Chao Ge[1], Shuangyue Cui[1], Leilei Zhang[1] & Xutang Tao [1]

Organic cocrystals possess valuable properties owing to the synergistic effect of the individual components. However, the growth of molecular cocrystals is still in its primary stage. Here we develop a microspacing in-air sublimation method to grow organic cocrystals, and furthermore to realize morphology control on them, which is essential for structure–property relations. A series of polycyclic aromatic hydrocarbon (PAH)–1,2,4,5-tetracyanobenzene (TCNB) complexes cocrystals are grown directly on the substrate, with the morphology tunable from 1D needle-like to 2D plate-like on demand. Spatially resolved photoluminescence spectra analyses on different cocrystals display morphology dependent and anisotropic optical waveguiding properties. In situ observation and energy calculations of the crystallization processes reveal the formation mechanism being from a competition between growth kinetics-defined crystal habit and the thermodynamics driving force. This growth technique may serve the future demand for tunable morphology organic cocrystals in different functional applications.

[1] State Key Laboratory of Crystal Materials, Shandong University, 250100 Jinan, Shandong, People's Republic of China. Correspondence and requests for materials should be addressed to Y.L. (email: liuyangicm@sdu.edu.cn) or to X.T. (email: txt@sdu.edu.cn)

Organic semiconductors have established their roles as active components in numerous emerging fields of both industrial practices and fundamental researches[1,2]. Being molecular solids, the main optical and electronic functionalities of organic semiconductors are overwhelmingly derived from their molecular structures, because when compared with the intramolecular covalent bonds, the intermolecular Van der Waals forces are much weaker and less dominant in determining the intrinsic properties[3–5]. On the other hand, an increasing number of studies have proved that the collective behavior of molecules also depends to a great extent on how they are arranged with their neighbors. The study about molecular crystal engineering adds another dimension to tune the properties of the orderly arranged organic molecules[5]. Cocrystals, which are composed of two or more different neutral molecules in a same crystalline phase, provide a versatile approach to expand the utility of one molecule by relating it with other molecules of interest in ordered solids[6]. For example, in pharmaceutics, people use cocrystallization of active pharmaceutical ingredients and cocrystal formers to improve the solubility or ease of formulation of drugs[7–9]. Moreover, owing to the synergistic effects of the individual components in cocrystals, new and valuable properties emerge not just from the activity of individual molecules but from their collective arrangement. Thus, cocrystals have attracted extensive attentions in many fields such as organic semiconductors[10,11], nonlinear optics[12], ferroelectrics[13,14], energy transfer systems[15,16], and energetic materials[17,18]. The co-assembly of multicomponents has been thoroughly investigated in the context of supramolecular chemistry and molecular crystal engineering, where normally the halogen bonds, hydrogen bonds, π–π interactions, charge-transfer (CT) interactions or other noncovalent interactions are in charge of the assembly forces. There is also a great deal of reports about the design strategy and applications of multicomponents molecular complexes[6].

Along with the increasing number of experimental case studies of cocrystal complexes, the demand for more effective growth method of cocrystals is also increasing[19]. Until now the most prevailing growth methods of cocrystal are based on solution-processed self-assembly via solution cooling, solvent evaporation, or diffusion of mixed solvents[6,20,21]. Mechanochemical synthesis by neat grinding or liquid-assisted grinding provides a versatile approach for pharmaceutical materials; but the low crystallinity and irregular morphology is not suitable for semiconducting applications[22,23]. In contrast, the cocrystals formed through vapor-based methods are often of high quality and clearly faceted shapes[24]. Significant improvements in physical property of semiconducting charge-transfer cocrystals have been made by adopting physical vapor transport (PVT) technique[25–27]. When considering the ease of integration into devices, such cocrystals with dimensions of nano- to micrometers grown directly on the substrates were preferred in most of the applications. Whereas, the morphology of the grown cocrystals, which are inherited from their intrinsic crystal habit[28], is hardly to be altered. Recently we developed a microspacing in-air sublimation (MAS) to grow organic crystals directly on the substrate[29]. The method can be conducted totally in air, avoiding costly vacuum system and time-consuming procedures in common PVT.

Herein, we employ MAS to grow cocrystals of a series of polycyclic aromatic hydrocarbon (PAH)–acceptor complexes and hydrogen-bonded pharmaceutical cocrystals. Because of the microspacing distance between the source and growth position in MAS, we realize morphology control over the cocrystals of different CT complexes. By precise manipulation of the starting materials (mixing by grinding or blend without grinding), the cocrystals grown on the substrate can be endowed with one-dimensional (1D) or two-dimensional (2D) morphology on

demand. The different cocrystals show morphology-dependent and anisotropic optical waveguiding properties, which may fulfill different requirements in real applications. Through in situ monitoring the crystallization process we demonstrate how the different growth conditions of MAS affects the cocrystal morphologies.

## Results

**Cocrystal growth and characterization of 1D and 2D FTCs**. The MAS is applicable to grow a series of different PAH–1,2,4,5-tetracyanobenzene (TCNB) complexes, e.g., anthracene–TCNB, pyrene–TCNB, and fluoranthene–TCNB cocrystals. (Supplementary Fig. 1) TCNB-based cocrystals generally exhibit enhanced luminescence owing to the CT transition from π-conjugated electron donor to the TCNB electron acceptor, which have been studied as models of light-emitting and harvesting systems[16,30,31]. Moreover, the distinct luminescence color for TCNB-based cocrystals with respect to one-component crystals facilitates recognition of the formation of cocrystals. So considering that fluoranthene crystal possesses obvious blue fluorescence, firstly we selected fluoranthene–TCNB complex as a model to explore the practicability of this method in the control of cocrystal growth. Figure 1a, j schematize the MAS apparatus. Being similar with that for single-component crystal growth, it consists merely of a hot-stage and two silicon wafers as the bottom and top substrates. The powders of TCNB and fluoranthene (molar ratio 1:1, total mass ~0.1 mg) were located together on the bottom substrate. The target substrate covered the starting materials with a tiny space of 150 μm from the bottom substrate, separated by two small glass sleepers. The growth procedure was implemented by heating the bottom substrate at about 130 or 140 °C for 30 min. Fluoranthene–TCNB cocrystals (FTCs) with micrometer (μm)-scale in size and nanometer (nm)-scale in thickness formed on the lower surface of the top substrate (Fig. 1b, f).

The whole sublimation growth process was accomplished at ambient pressure in air. According to the equation of Rayleigh number (Ra)[32,33], which is associated with buoyancy-driven flow and is used to demarcate the vapor and heat transport regimes:

$$\mathrm{Ra} = \frac{g\beta\Delta T h^3}{\nu\kappa} \qquad (1)$$

where $g$ is acceleration due to gravity, $\beta$ is the thermal expansion coefficient, $\nu$ is the kinematic viscosity, $\kappa$ is the thermal diffusivity, $\Delta T$ is the temperature difference and $h$ is the spacing length between the source and deposition positions, a microspacing distance of 150 μm leads to a small Rayleigh number estimated to be $1.7 \times 10^{-7}$, because Ra is in direct proportional to the third power of the spacing distance. An efficient buoyancy-driven molecular flow vapor transport mode would be generated within the confined space between two substrates, which on one side facilitates the evaporation of the starting materials although the temperature is much lower than the melting point of TCNB (268 °C), and on the other hand protects the molecules from oxidation or decomposition. More detailed discussions about the stability of the organic materials during the sublimation process refer to our last work[29]. The identity and purity characterizations of the grown FTCs are shown in Supplementary Fig. 2.

More importantly, morphology control over the grown cocrystals can be realized by changing the growth conditions. As shown in Fig. 1a, a key factor is the way how the two component molecules are mixed and distributed on the bottom substrate. That is, when the two kinds of component molecules were just put closely together without grinding, the cocrystals grown at 130 °C show 1D needle-like shapes on the top substrate (Fig. 1a). Oppositely, when the two starting molecules were mixed

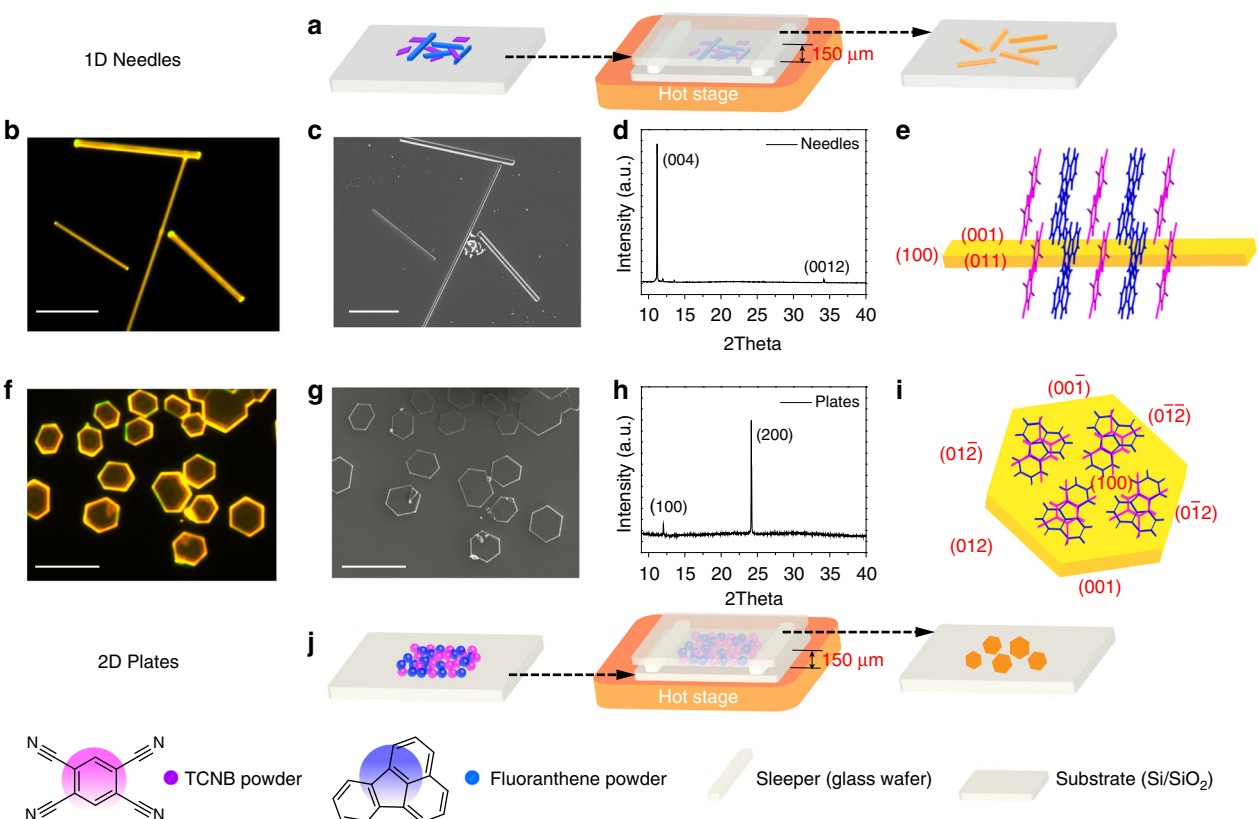

**Fig. 1** MAS growth of FTCs on the substrate and morphology characterization of as-grown FTCs. MAS apparatus for growth of 1D (**a**) and 2D (**j**) FTCs. Fluorescence microscopy images (**b**, **f**), SEM images (**c**, **g**), XRD patterns (**d**, **h**), and schematic models (**e**, **i**) of FTCs. (Scale bar: 50 μm for **b** and **c**, 25 μm for **f** and **g**). MAS microspacing in-air sublimation, FTCs Fluoranthene–TCNB cocrystals, SEM scanning electron microscopy, XRD X-ray diffraction

thoroughly by grinding and were heated at a higher temperature (140 °C) for 30 min, the grown cocrystals show 2D plate-like shapes (Fig. 1j). Fig. 1b–e, f–i show the characterization of the formed 1D needle-like crystals and 2D plate-like crystals, respectively. As shown in the fluorescence microscopy (Fig. 1b) and scanning electron microscopy (SEM) images (Fig. 1c), the length of the 1D crystals range from tens of to hundreds of μm, with the width of <5 μm and the thickness of ~1.2 μm (Supplementary Fig. 3a, b). The 2D plate-like crystals, correspondingly, show similar magnitude in length and width of 10–20 μm and ~550 nm in thickness (Supplementary Fig. 3c, d). Both the as-grown 1D and 2D cocrystals are well-dispersed on the substrate with smooth surface and high uniformity as revealed by the microscopy images.

Since cocrystals of fluoranthene–TCNB have not been studied in previous reports, the crystal structures of both 1D and 2D cocrystals were characterized by using X-ray diffraction (XRD). The single crystal XRD analysis reveals the two morphology cocrystals actually belonging to the same crystal structure of monoclinic $P2_1/c$ space group with $a = 7.362(5)$ Å, $b = 8.263(6)$ Å, $c = 31.29(2)$ Å, and $\beta = 90.250(14)°$. (The detailed structural information is provided in Supplementary Table 1.) As shown in Supplementary Fig. 4, the two planar constituent molecules stack parallelly and alternately along the $a$ axis, with a face-to-face distance of 3.5 Å, suggesting an intrinsic CT interaction between fluoranthene and TCNB. The 1D and 2D morphology cocrystals originated from differences in orientation of molecular packing on the substrate and the induced different growth habits. As shown in the XRD patterns of the respective 1D and 2D cocrystals as-grown on the substrate (Fig. 1d, h), only diffraction

peaks corresponding to (00l) or (h00) diffractions of the cocrystals appear. Based on the $2\theta$ scan, face-indexing of the grown crystals (Supplementary Fig. 4c) and growth morphology modeling based on the attachment energies by using the material studio package (Supplementary Fig. 4d)[34–36], the 1D cocrystals were formed via orientational growth of the complex molecules on the substrate with (001) face parallel to the substrate surface and $a$-direction parallel to the longitudinal direction of the needle-like crystals (Fig. 1e); The 2D cocrystals, on the other hand, were the equilibrium shape with minimum total surface energy (Supplementary Fig. 4f) and formed via orientational growth with (100) face parallel to the substrate surface and $a$-direction perpendicular to the surface (Fig. 1i). Other facets appeared on the 1D and 2D cocrystals are also labeled as shown in Fig. 1e, i according to the interfacial angle. From a molecular viewpoint, the planar fluoranthene and TCNB molecules were standing perpendicularly or lying parallelly to the substrate in the 1D and 2D cocrystals, respectively.

The cocrystals show intense yellow fluorescence (Fig. 1b, f) and the photoluminescence (PL) spectrum (Supplementary Fig. 5a) of the cocrystals shows red-shifts of 100 and 210 nm, as compared with that of the two individual constituent molecules fluoranthene and TCNB, respectively (from 455 and 345 to 555 nm). This dramatic red-shift is attributed to the CT interaction between fluoranthene and TCNB, which is further corroborated by the absorption spectra of the cocrystals and the two constituent molecules. As shown in Supplementary Fig. 5c, the new absorption band appearing around $\lambda_{abs} = 509$ nm is ascribed to characteristic CT bands. The PL quantum yield ($\Phi_f$) of the cocrystal is determined to be 74%, which is much higher than that

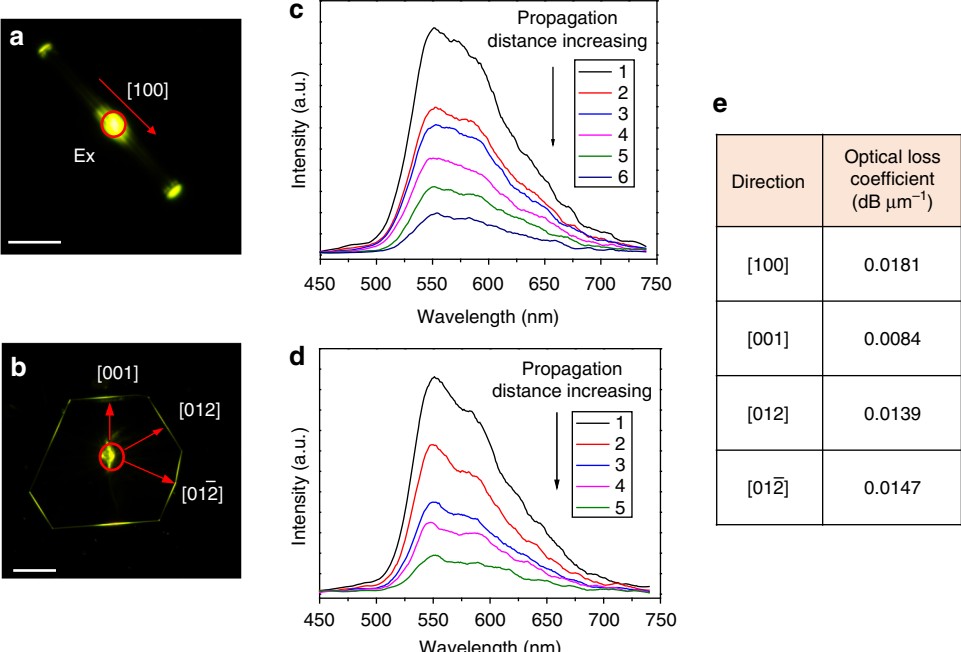

**Fig. 2** Anisotropic optical waveguiding effect in 1D and 2D FTCs. Fluorescence microscopy images (**a**, **c**, red circles represent the exciting light spot) and spatially resolved PL spectra collected from the tip or edge of FTCs when the excitation spot was moved on 1D FTC (**b**) and 2D FTC (**d**). **e** Optical propagation loss coefficient in different crystallographic directions. (Scale bar: 50 μm). FTC fluoranthene–TCNB cocrystals, PL photoluminescence

of TCNB ($\Phi_f = 4\%$) and fluoranthene ($\Phi_f = 20\%$)[37], being a record value for TCNB-based complex reported so far[16,38–41]. Correspondingly, the PL lifetime of cocrystals is also much longer than the monomers (80 ns versus 8.08 ns for TCNB and 43 ns for fluoranthene) (Supplementary Fig. 5b).

**Morphology-dependent optical waveguide of 1D and 2D FTCs.** Employing the as-grown 1D and 2D FTCs as active optical waveguide resonators, we investigate the structure–property relationship of the distinctive crystalline microstructures. The morphology difference of the cocrystals indeed endow them with different optical confinement behaviors. As presented in Fig. 2a, when being excited by a 375 nm laser focused at the middle of a 1D cocrystal, the photons propagate along two predominant transmission directions that confined in one dimension. Except the excitation spot, only bright out-coupled light beams from the two tips of the needle-like cocrystal were observed. Figure 2b shows the spatially resolved PL spectra acquired from the tip when moving the laser excitation position to increase the light propagation distance, from which the optical-loss coefficient of the waveguiding along 1D direction ([100]) was determined to be 0.0181 dB μm$^{-1}$ (Fig. 2e and Supplementary Fig. 6b). For the plate-like 2D FTCs, being excited at the center part generates out-coupled light beams from all the six edges of the plate along three crystallographic directions (Fig. 2c). Figure 2d and Supplementary Fig. 6d–f illustrate the corresponding spatially resolved PL spectra detected from the 2D cocrystal edge by changing the optical propagation distance along [001], [012], and [01$\bar{2}$] directions. Correspondingly, the in-plane optical propagation loss coefficient along [001], [012] and [01$\bar{2}$] direction is calculated to be 0.0084 dB μm$^{-1}$, 0.0139 dB μm$^{-1}$, and 0.0147 dB μm$^{-1}$, respectively. (Fig. 2e and Supplementary Fig. 6g–i). These loss coefficients are much lower than the value for other organic crystals[42,43]. We attribute the good waveguiding behavior to two important factors. First, the large red-shift of cocrystal emission induced by CT interaction between fluoranthene and TCNB

reduces the overlap with its absorption, which effectively diminishes the reabsorption during light propagation. Second, the smooth surface and high single crystallinity minimizes the scattering caused optical-loss, which also proves the great usability of our method for cocrystal growth. Additionally, the asymmetric light propagation along a same crystallographic direction in the 2D cocrystal was not observed[44], while it indeed shows an anisotropic waveguiding behavior. That is, the optical-loss coefficient along [001] direction is the smallest and those along crystallographically relevant [012] and [01$\bar{2}$] directions are similar and largest. The anisotropic molecular packing mode caused optical transition dipole was believed to determine the final optical propagation efficiency[42,44]. Supplementary Fig. 7 represents the anisotropic organic molecular packing in different crystallographic planes and along different directions within 2D FTCs. Considering the diversity of application scenarios for active organic crystals, the morphology controllability from 1D to 2D is, therefore, able to fulfill different preferences or requirements in different device applications.

**One-dimensional (1D) and 2D cocrystal formation process and mechanism.** The growth mechanism resulted from microspacing is found to be essential for the morphology control over the grown cocrystals in different implement conditions. To probe how the sublimation conditions determine the morphology of the grown cocrystals, we investigated the crystal formation process by in situ observation of the growth process on a hot-stage fluorescence microscope. Figure 3 represents the sequential scenarios emerged on both the top and bottom substrates during the growth process of the 1D and 2D cocrystals, respectively. To grow 1D cocrystals, the two starting materials were just put on the source substrate in the forms of microcrystals without grinding (Fig. 3n). Blue emission from fluoranthene crystals and weak fluorescence from TCNB could be witnessed before heating. As the temperature increased to 90 °C, fluoranthene with lower melting point ($T_m = 110$ °C) sublimated firstly. So fluoranthene

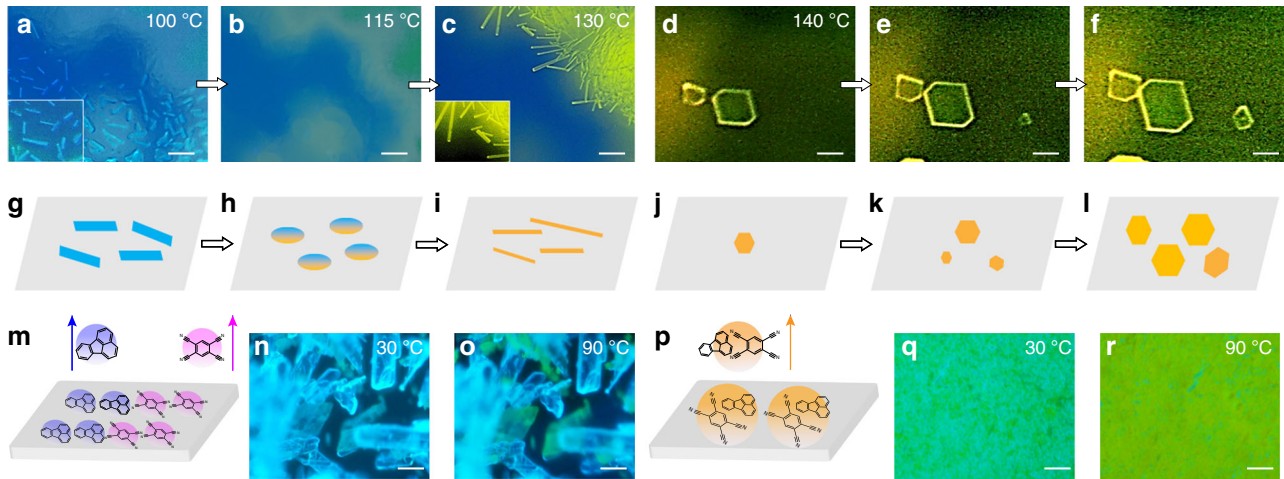

**Fig. 3** Real-time observation of crystal formation process of 1D and 2D FTCs via hot-stage fluorescence microscope. The process observed on the top quartz substrates after heating to 100 °C (**a**), 115 °C (**b**), 130 °C (**c**), 140 °C for 5 min (**d**), 10 min (**e**), and 15 min (**f**), inset is the enlarged view. The images of starting materials on the bottom substrates before (**n**, **q**) and after (**o**, **r**) heating. The corresponding schematic outline of crystallization mechanism in the MAS method for 1D (**g–i**, **m**) and 2D (**j–l**, **p**) FTCs. (Scale bar: 20 μm for **d–f**, 100 μm for others). FTCs fluoranthene–TCNB cocrystals

crystals formed firstly on the down surface of the top substrate (Fig. 3a). In the meantime, small portion of the sublimed fluoranthene reacted with TCNB, making parts of its surface conversion into yellow (Fig. 3o). When the temperature reached 115 °C, which is higher than the melting point of fluoranthene, the formed fluoranthene crystals on the top substrate melt into liquid droplets (Fig. 3b). And when the temperature reached 130 °C, TCNB started to sublimate; cocrystals crystalized from the melt with 1D needle-like shapes and yellow emissions. (Fig. 3c) For the growth of 2D plate-like cocrystals, the two starting molecules were thoroughly mixed through grinding, which already showed yellowish-green before heating due to the formation of fluoranthene–TCNB complex (Fig. 3q). After being heated to 90 °C, the starting materials on the bottom substrate were almost totally converted into yellow emissive complex (Fig. 3r). In this case, the materials started to sublimate until the temperature increased to 140 °C, at which point the two kinds of molecules sublimated synchronously as complex, thus no fluoranthene crystals formed on the top substrate as precursors. The 2D plate-like cocrystals formed directly on the substrate and grown bigger as the sublimation time went on (Fig. 3d–f).

Figure 3g–l schematized the crystallization mechanism based on the in situ observation. In the growth of 1D cocrystals, the starting materials blended without grinding (Fig. 3m) resulted a two-step sublimation. The firstly grown fluoranthene crystals (Fig. 3g) melted anew as the temperature rising and met with the latter sublimated TCNB (Fig. 3h). The donor and acceptor molecules formed CT complex and then the cocrystals nucleated and grew from the liquid melt, due to a higher melting point of the cocrystal (217 °C, Supplementary Fig. 8) than that of fluoranthene. The crystallization from liquid melt is more likely to be a spontaneous process, thus resulting crystal habit to be defined by the attachment energy between lattice layers just like those grown from solutions[41,45–47]. (FTCs grown from solution show 1D morphology. Supplementary Fig. 9) The strong CT intermolecular interactions between the two component molecules lead to the favored growth direction along the CT interaction, i.e., *a* axis. (Fig. 3i) Thus the high growth speed along [100] direction produced 1D needle-like FTCs, which is consistent with the morphology for other PAH–TCNB cocrystals grown from solutions studied previously[45,46]. For the growth of 2D cocrystals, the molecular complex of fluoranthene–TCNB

formed during the grinding and preheating stage before sublimation (Fig. 3p). The complex sublimated simultaneously at a relative higher temperature (140 °C). Two-dimensional (2D) cocrystals formed on the substrate directly, with the (100) plane as the domain face, where the plannar molecules lie parallelly to the substrate surface (Fig. 3l). From the view of thermodynamics, the crystallization driving force can be expressed as the difference in the chemical potential between molecules in the ordered and disordered phase ($\Delta\mu$). The growth morphology is related to the relationship between $\Delta\mu$ and the kinetic barrier of each of the crystal facets ($\Delta G$); and $\Delta G$ is inversely proportional to the surface energy ($\gamma$)[34]. According to the surface energies for FTCs calculated by the material studio[34,35] (Supplementary Table 1), $\gamma_{(100)} > \gamma_{(012)} > \gamma_{(011)} > \gamma_{(002)}$, $\Delta G_{(100)}$ has the lowest barrier. The adjacency of (100) plane to the substrate surface would be most favored thermodynamically. At a higher temperature, $\Delta\mu$ is high enough to overcome the growth barrier of $\Delta G_{(012)}$, which capacitates the growth of (012), (01$\bar{2}$), (0$\bar{1}$2), and (0$\bar{1}\bar{2}$) planes and results in 2D plate-like shapes. Thus, the morphology control is a competition effect between kinetics-defined crystal habit and the thermodynamics driving force.

**Generality of cocrystals growth by MAS.** The morphology control over 1D to 2D cocrystals is found to be also appropriate for other PAH–TCNB CT complex, e.g., anthracene–TCNB and pyrene–TCNB cocrystals. As shown in Fig. 4, blend without grinding of the two starting materials and growth at relative low temperature (140 °C for anthracene–TCNB and 130 °C for pyrene–TCNB) produced 1D needle-like cocrystals. Oppositely, thoroughly mixing of the two starting molecules through grinding and growth at relative higher temperature (150 °C for anthracene–TCNB and 140 °C for pyrene–TCNB) is more conducive to obtain 2D cocrystals.

Besides the PAH–TCNB complexes, we also tested other cocrystals to check the generality of the method, including PAH–haloperfluoroarene systems and even hydrogen-bonded pharmaceutical cocrystals. A typical carbamazepine–saccharin[48] pharmaceutical cocrystal, which was conventionally prepared from solution, was employed in the MAS growth. As shown in Fig. 5b, plate-shaped carbamazepine–saccharin cocrystal with size of about 10–30 μm was grown on the substrate. Its XRD pattern is

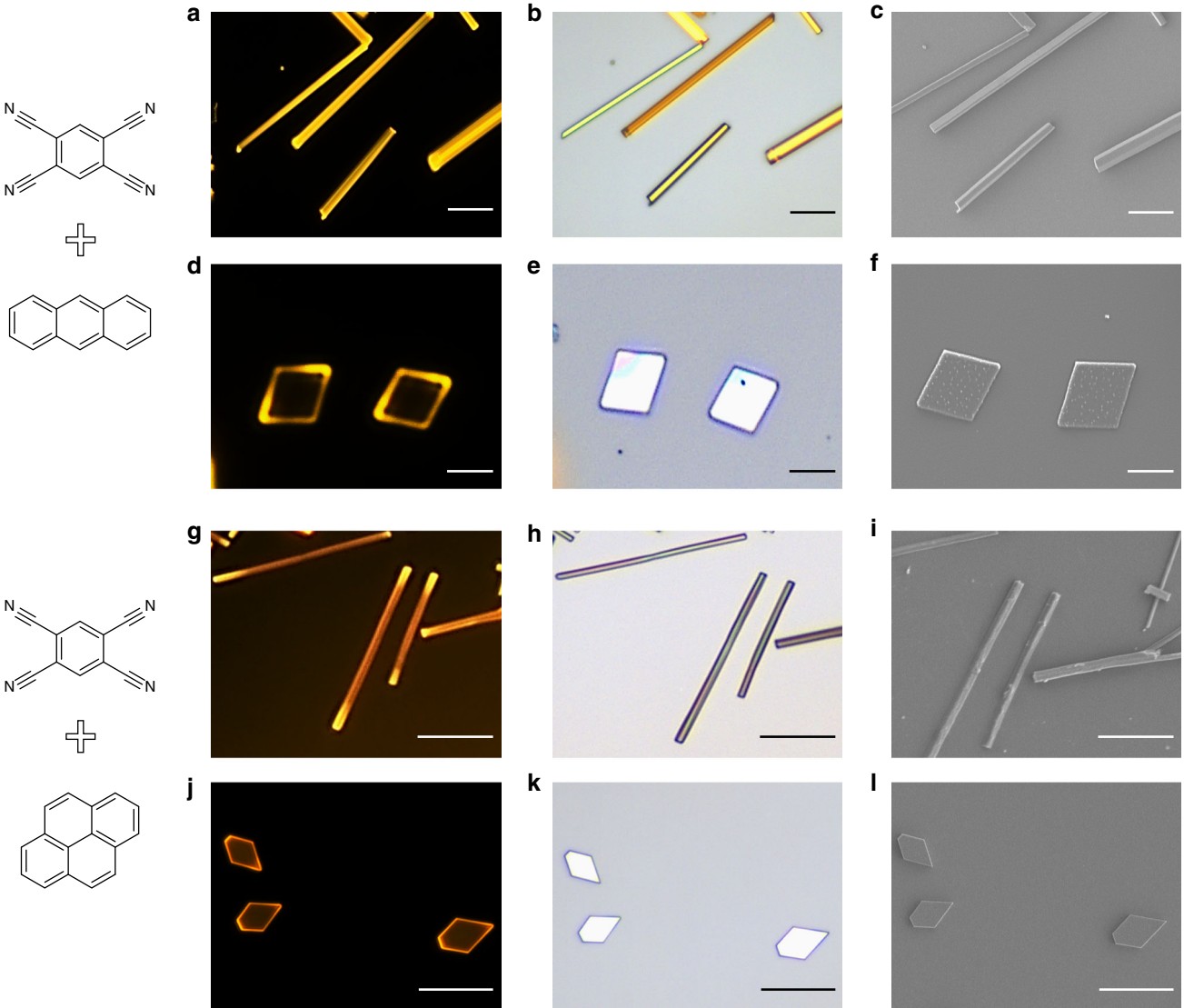

**Fig. 4** One-dimensional (1D) and two-dimensional (2D) cocrystals growth of other PAH–TCNB complexes. Fluorescence microscope images (**a,d**, **g**, **j**), microscope images (**b**, **e**, **h**, **k**) and SEM images (**e**, **f**, **i**, **l**) of as-grown anthracene–TCNB (**a–f**) and pyrene–TCNB (**g–l**) cocrystals. (Scale bar: 5 μm for **d–f**, 20 μm for others). PAH–TCNB polycyclic aromatic hydrocarbon (PAH)–1,2,4,5-tetracyanobenzene (TCNB) complexes, SEM scanning electron microscopy,

indexed to a triclinic $P\bar{1}$ polymorph, and indicates an oriented growth with $c$-axis perpendicular to the substrate surface (Fig. 5c). Another pharmaceutical cocrystal carbamazepine–nicotinamide[49] is also applicable to the method, for which needle-like cocrystal with oriented growth was obtained (Supplementary Fig. 10). For the PAH–haloperfluoroarene systems, pyrene–1,3,5-trifluoro-2,4,6-triiodobenzene[50] cocrystal was successfully grown on the substrate, with morphology control from 1D to 2D by changing the growth conditions likewise (Fig. 5d–h and Supplementary Fig. 11). Whereas for the anthracene–octafluoronaphthalene[51] complex, growth at relative lower temperature did produce 1D needle-like cocrystal; growth at relative higher temperature (higher than 120 °C) obtained 2D plated anthracene crystal, but not cocrystal (Supplementary Fig. 12). The failure of morphology control over anthracene–octafluoronaphthalene cocrystal is attributed to the incompatibility between the vapor pressures of the two components, where the low melting point (87-88 °C) and high vapor pressure of octafluoronaphthalene make it volatilize from the cocrystal with temperature increasing (Supplementary Fig. 13). The results demonstrate that our MAS method

has a broad universality in cocrystals growth that even hydrogen-bonded pharmaceutical cocrystals are applicable. Morphology control at high temperature should be aware of the compatibility between the vapor pressures of the constituent molecules.

**Two-dimensional (2D) FTCs with sixfold symmetric patterns.** Interestingly, for the 2D plate-like cocrystals, if the growth temperature is between 130 °C and 140 °C, sixfold symmetric patterns occurred in the center part of the cocrystals. As shown in Fig. 6a, c, and d, six round craters with diameter of ~3 μm correspond directly with the six facets of each 2D plate-like crystal. The patterns only formed on the bottom face of the crystal, that is, the adjacent plane to the substrate, so from the SEM image of the front side we cannot see any patterns (Fig. 6b), while the back-side image shows clear sixfold patterns (Fig. 6d). Currently we are not clear about the formation mechanism of these sixfold patterns. Previously several researchers have investigated the growth of cocrystal microtubes with hollow structures, in which they attribute the formation mechanism to be the coalescence growth

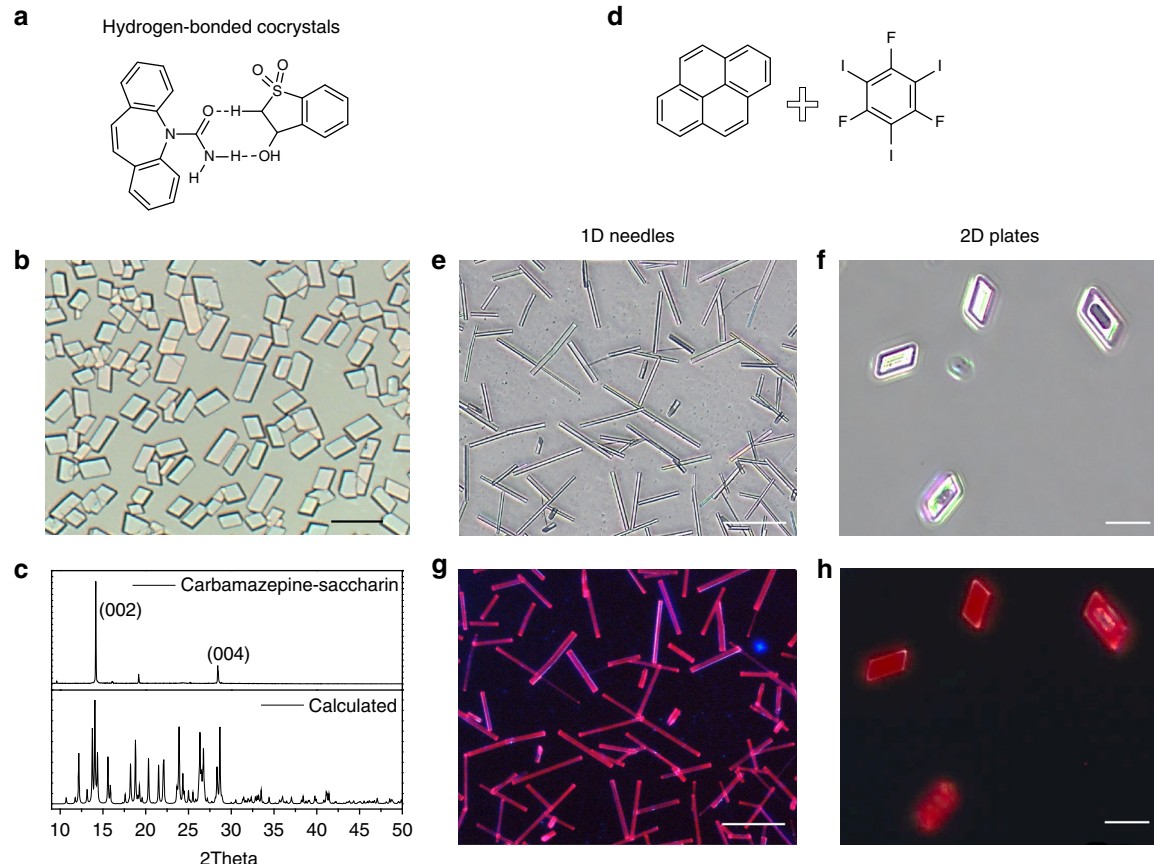

**Fig. 5** Generality of MAS in growth of other cocrystals. Chemical structures of carbamazepine–saccharin (**a**) and pyrene–TIFB (**d**). Microscope image (**b**) and XRD patterns (**c**) of the as-grown carbamazepine–saccharin cocrystals. Microscope images (**e**, **f**) and fluorescence microscope images (**g**, **h**) of 1D needle-like and 2D plate-like pyrene–TIFB cocrystals. (Scale bar: 50 μm for **b**, **e**, and **f**, 20 μm for **f** and **h**). MAS microspacing in-air sublimation, XRD X-ray diffraction

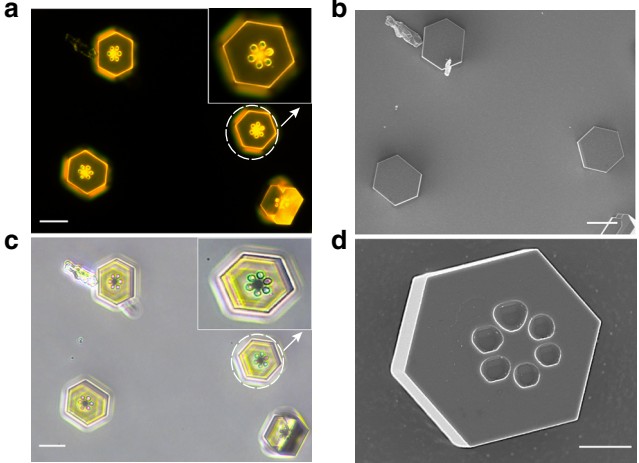

**Fig. 6** Morphology characterization of 2D FTCs with sixfold symmetric patterns. Fluorescence microscopy image (**a**) and microscope image (**c**) of the back-side (inset is the enlarged view), SEM images of front side (**b**) and back-side (**d**). (Scale bar: 20 μm for **a**–**c**, 5 μm for **d**). FTCs fluoranthene–TCNB cocrystals

of neighboring random and isolated elements, and solvent etching or dissolution of inner part molecules by poor solvent in solution[16,52]. After carefully comparative checking of their case with our growth process, the aforementioned mechanisms are found to be not applicable to our case. The appearance of the patterns may be related to the initial stage of the growth process during developing of the six crystalline facets. Since there have been no reports about growth of organic microcrystals embedded with such sixfold symmetric patterns, and these optoelectronic crystals with concomitant sub-ensembles can be served as optical scattering modules, future analysis about the formation mechanism and exploration on their peculiar properties in electronic and optical modulations are required.

## Discussion

In summary, considering the fact that till now neither can we predict the formation of cocrystals, nor have we any ways to control the morphology of the grown cocrystals, developing efficient and versatile growth methods to screen the feasibility from countless potential partner molecules, and furtherly to regulate the crystalline habit and orientation, is thus meaningful for the real applications of molecular cocrystals. Here, we developed a method to grow cocrystals by MAS. A series of PAH–TCNB cocrystals were successfully grown directly on the substrate. The method has a broad universality in cocrystals growth that even hydrogen-bonded pharmaceutical cocrystals are applicable. More importantly, the morphology of the cocrystals can be facilely controlled by manipulation of the mixing condition of the component molecules and growth temperature. 1D needle-like and 2D plate-like cocrystals with high crystallinity and anisotropic optical waveguiding property can be grown on demand, in which the constituent molecules are actually standing and lying on the substrate, respectively. In situ observation of the crystallization processes revealed two-step sublimation for 1D

growth while synchronous sublimation for 2D growth. Because most application scenarios in optoelectronic devices have different morphology preference or requirements for the integrated cocrystals, this growth technique for molecular cocrystals, which is provided with morphology control ability, will pave the way for potential applications of molecular complex materials.

## Methods

**Materials**. Fluoranthene (98%), anthrance (99%), pyrene (99%), coronene (99%) were purchased from Aladdin. 1,2,4,5-tetracyanobenzene (TCNB, 98%) was purchased from TCI.

**Growth of single crystals**. The mass of starting materials is 0.1 mg (molar ratio 1:1, mixed without grinding for needle shaped cocrystals and mixed by grinding for plate-shaped cocrystals), the heating rate is 50 °C/min and the growth period is 30 min. The detailed growth temperature are as follows: fluoranthene−TCNB (130 °C for needle shaped and 140 °C for plate shape), anthracene−TCNB (170 °C for needle shaped and 180 °C for plate shape), pyrene−TCNB (130 °C for needle shaped and 140 °C for plate shape), carbamazepine−saccharin (160 °C), carbamazepine−nicotinamide (140 °C), pyrene−TIFB (120 °C for needle shaped and 140 °C for plate shape, with the space distance of 1 mm), anthracene−OFN (100 °C).) Selection of optimal experimental parameters for different materials, including the space $h$, growth temperature and the mixing and distribution state of starting materials, has essential effect on the cocrystal morphology and crystallinity. For example, an improper space $h$ would result poor quality of the grown cocrystals. (Supplementary Fig. 14)

**Hot-stage fluorescence microscope**. The hot-stage fluorescence microscope system consists of a Linkam THMS600 heating stage and an optical microscope (PSM-1000) equipped with a 365 nm UV LED.

**X-ray diffraction**. The powder X-ray diffraction was performed on a Bruker-AXS D8 ADVANCE X-ray diffractometer equipped with a diffracted beam monochromator set for Cu Kα radiation ($\lambda = 1.54056$ Å) in the range of 9–40°($2\theta$), with a step size of 0.08° and a step time of 0.2 s at room temperature. Crystal structure analysis was carried out with a Bruker SMART APEX-II equipped with a CCD area-detector diffractometer at 298 K using graphite monochromated $Mo_{Ka}$ radiation ($\lambda = 0.71073$ Å) with the w scan method. The structure was solved by direct methods and refined by the full-matrix least-squares technique on $F^2$ using SHELX programs.

**Fluorescence microscope measurement**. Fluorescence images were obtained using a Nikon Ti-U Inverted Microscope System equipped with a Nikon C-SHG 1 mercury lamp.

**PL spectra measurement**. Steady-state fluorescence measurements were performed with an Edinburgh Instruments FLS980 spectrometer. Fluorescence quantum yields were determined with the FLS980 spectrometer using optically dense samples in an integrating sphere. Fluorescence lifetimes were measured by time correlated single-photon counting (TCSPC) at the FLS980 spectrometer using pulsed laser diodes at 375 nm (EPL-375) as excitation source. Micro-area photoluminescence (μ-PL) spectra for the needle-shape and plate-shape single crystal were performed with FLS980 spectrometer equipped with a Nikon Ni-U Inverted Microscope. To measure the PL spectra of individual microcrystals, the cocrystal was excited locally with a 375 nm (EPL-375) laser focused down to the diffraction limit through an objective (Nikon CFLU Plan, 50×, N.A. = 0.8).

**Absorption spectra measurement**. The absorption spectroscopy was carried out using a conventional UV/Vis spectrometer (Hitachi U-4100) equipped with an integrating sphere over the spectral range 200–800 nm.

**Electron microscopy measurement**. SEM was obtained by a Hitachi S-4800 ultrahigh resolution (UHR) field emission (FE) scanning electron microscope and the pictures were taken at an accelerating voltage of 5.0 kV. Micrococrystals were prepared on a carbon-covered copper grid by MAS at ambient pressure. The thickness and surface morphology of crystals were measured by AFM at ambient conditions with a using a Digital Veeco Instruments atomic force microscope operating in the tapping mode.

**Thermal analysis**. The differential scanning calorimetry (DSC) measurement were implemented on a NETZSCH DSC 200F3 analyzer with a heating rate of 10 °C/min.

**Theoretical calculations**. The morphologies and energy calculations of FTCs were performed by the Material Studio software. The molecular structure was firstly optimized based on the experimental crystal structure using the Build Bonds and the energy calculations were performed using the Forcite and Morphology modules.

## Data availability

The data that support the findings of this study are available from the authors on reasonable request. The X-ray crystallographic coordinates for structure reported in this study have been deposited at the Cambridge Crystallographic Data Center (CCDC), under deposition number 1864519. These data can be obtained free of charge from The Cambridge Crystallographic Data Center via www.ccdc.cam.ac.uk/data_request/cif.

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

## Acknowledgements

We acknowledge support from the National Natural Science Foundation of China (Grant Nos. 21772115, 51227002, 51272129), National key Research and Development Program of China (Grant No. 2016YFB1102201), and the 111 Project 2.0 in China (Grant No. PB2018013). Y.L. thanks the support from Qilu (Zhongying) Young Scholars. We thank Hongkai Ren and Longfei Xiao for help with μ-PL measurements.

## Author contributions

Y.L. and X.T. had the idea for and designed the experiments and supervised the work, X.Y. did the major experiments and analyses, C.G. contributed to the thermal analysis. S.C. and L.Z. prepared the samples, Q.H. and Q.G., helped the analyses of the crystallization process and the TEM data, X.Y. and Y.L. wrote the manuscript, X.T. provided major revisions of the manuscript.

## Additional information

**Competing interests:** The authors declare no competing interests.

