## [Peer Review File · Nature Communications]

Reviewers' comments:

Reviewer #1 (Remarks to the Author):

The manuscript, "1D Versus 2D Cocrystals Growth via Microspacing In-Air Sublimation" describes a very simple and efficient method to crystallize the same binary organic compounds in two different morphology. As authors correctly stated the physical properties depend not only on the choice of molecules but also on the morphology of thin films. These are interesting experimental results. Few points may be discussed in more details.

Authors claim that the crystallization of co-crystals is "is still in its primary stage" and cited the ref. [17]. Just during the last few years, some detailed papers appeared in *Crystal Growth and Design* reporting charge transfer compounds crystallization. Charge transfer compounds are co-crystals, therefore, authors should refer to these papers and including these papers in their discussion.

It is also difficult to agree with authors claim that "Unfortunately, little advances have been achieved in this respect, wherein a complex physical vapor transport (PVT) conducted in vacuum or inert gas was commonly adopted" PVT in an inert gas is not a complex technique and the progress in physical property improvements of organic semiconductors made from charge transfer compounds are significant.

The microspacing in-air sublimation (MAS) is definitely a very simple method, however, due to possible oxidation of some air sensitive molecules it has specific technological limitation only very shortly discussed in this paper.

In the experiment description, authors write the bottom and top substrate was separated by "a tiny space of 150 μm ". What was the average grain dimensions of used powder in both experiments when a grinded or a mixture of two powder is used? How the powder was "simultaneously located" on the top substrate?

Figure 1 is confusing. The section 1a suggest that these two compounds are widely separated from each other. Therefore, one or both components need to travel long horizontal distances to form a homogeneous co-crystals with constant stoichiometry. Is it the case? Authors' commentary on this would be helpful.

Authors claim that a convection flow should be observed according to calculated Rayleigh number. However, the values of these number are not given. Also, the value of Rayleigh number limiting laminar flow from convection could be useful for process understanding.

In line 121, authors write that crystallization temperature was 140 oC. However, in line 93 the bottom substrate temperature was 130 oC. It seems that it is hard to decide if the morphology of powders or the substrate temperature was responsible for needle or platelet growth.

The conclusion of this paper could be found at the end of the paper and it is: "In situ observation of the crystallization processes revealed two-step sublimation for 1D growth while synchronous sublimation for 2D growth" Experimental results from these paper seems to confirm this conclusion. Especially, detailed described growth mechanism by sequential evaporation of individual components for 1D morphology and simultaneous evaporation of components for 2 D morphology is the leading thought constructing this paper.

Reviewer #2 (Remarks to the Author):

Referee Report

Authors: Ye, Liu, Han, Ge, Cui, Zhang, Tao

Title: 1D Versus 2D Cocrystals Growth via Microspacing in In-Air Sublimation

I am reluctant to use the relatively noncommittal adjective "intriguing" to describe this manuscript. However, its use is definitely meant in the positive sense. As a long time player in the discipline of organic crystals, and with some experience in the kind of charge transfer complexes described here I found the results presented both surprising, and yet frankly questioned why this effect had not been previously observed.

The authors present rather convincing experimental evidence to support their claims. While I had doubts about the general application of this sublimation phenomenon to other systems, the two final examples of pyrene-TCNB and anthracene-TCNB did serve to allay some of those doubts. Nevertheless, I found myself asking the question of whether any other acceptor would behave the same way. Some other questions I found myself asking: Does there have to be some compatibility between the vapor pressures of the two components? What would be the result of an experiment in which, say, the two different preparations of the two components were arranged in opposite corners of a square arrangement? Would a sample of mixed crystal habits be obtained? How strictly must the experimental parameters be maintained in order to maintain control over the product? As noted above: How general is this phenomenon? Is it limited to pi charge transfer complexes? If so, do the authors have any insight into which donors and which acceptors? Would hydrogen bonding co-crystals be suitable candidates for such experiments? Are there some lower limits on the vapor pressure of one or both of the solid components?

As noted above, most of the questions here fall into the "intriguing" category, rather than criticism; nevertheless, I felt that I was looking for answers. On the other hand, the results presented here certainly suggest new areas to explore and the paper would serve as a catalyst for those studies. May TCNB is not the only pi-acceptor that would exhibit this controlled habit change under different sublimation conditions. In that sense, the paper is worthy of publication – as a catalyst for further work in an area that has not yet been explored.

While the science is intriguing, much of the writing needs improvement, as noted below.

Some more detailed comments follow:

L14 The abstract seems to indicate that this area of cocrystals – in particular molecular pi complexes is new. It has been recognized for nearly a century [e.g. Paul Pfeiffer's *Organische Molekülverbindungen*, Verlag von Ferdinand Enke, Stuttgart, 1927 and R.S. Mulliken's pioneering work in the 1950's and 1960's. to say nothing of subsequent developments, say in the area of organic conductors and superconductors]

L17 Morphology control is indeed an important aspect of structure property relations. It is very much (but of course, not exclusively) a matter of the kinetics of crystal growth. Those two aspects are essentially ignored by the authors.

L25 What is the meaning of "structural inheritance"?

L31 Being molecular solids...

L37 This is an awkward sentence

The "sentence" beginning on line 38 "Cocrystals..." is not a sentence.

L43 There are better reviews than Reference 7. See, e.g. those by Zaworotko and/or Almarrson, even since 2004, but there are more recent ones as well.

L46 Again, an awkward phrase

Lines 48-53 Again, the writing here needs some serious improvement. In fact, all of the text of lines 14-53 should be the subject of some serious editing for language and construction.

Line 59 "resulted" is not the right word

Line 62 "When considering the ease of integration into devices..." Why

Line 64 Again, the use of "little advances" is awkward English

Line 66-67 What is "their intrinsic molecular packing symmetry"? "could be hardly affected through man-made regulations" also doesn't make much sense.

Line 72 "novel growth mechanism". The growth mechanism is not novel. The experimental technique is novel.

Line 74 "different implement techniques" is awkward, and poor English.
Ditto line 76 "with grinding mix or not"

Line 80 "monitoring of the crystallization process". Is it growth mechanism or growth conditions?

L85 Why. "firstly"? Specifically, what was the reasoning behind the choice of this system? Was it based on some chemical reasoning, some observation, some literature reference, some other experience in the laboratory? Was it based on a survey of many other compounds and combinations of compounds? How would a future worker have a basis for using this technique to influence crystal habit? That is a crucial lesson that is missing from this paper.

L90 The "simultaneously located" does not indicate of mixed or separate.

L95 Change "down" to "lower".

◇ And the question that I asked when I got to this point was, "Does it depend on similar vapor pressures for the components? I never received an answer to that question.

Figure 1 How many times was this experiment performed? Upon achieving success, does it always work under the conditions described? How critical is the space h? Anybody involved in growing crystals as part of their experimental activities knows that not every crystallization works, even ones that have been done many times. Something about the history and the success rate needs to be included here.

Line 105 How do the authors know that this equation applies under the conditions of this system?
Line 114 This sentence is awkward.

Line 118 These instructions are really not very clear.

Line 121 Why "peculiar"?

Line 123 "As we can see" is inappropriate style.

Line 131, and a few other places below this. The first "n" of fluoranthene is missing.

Line 138 The face-to-face distance of 3.5Å is actually quite normal for complexes of this sort.

Line 147. I don't understand why this is labeled the (002) face. It is true that for P21/c the 001 reflection is absent by symmetry, but the face is still there.

Lines 161-162 should read "...and the two individual constituent molecules". An appropriate reference should be added.

Line 250 Why is the crystallization from the liquid melt more likely to be a spontaneous process.

Line 251 What exactly is the meaning of "inherited from the intrinsic crystal structure"/

Line 264 "...of each of the crystal faces..."

Line 274 Delete "I" after Figure 4

Line 303 What are "active electric disturbance" and "optical scattering modules".

Line 305 Change "much needed" top "required".

Line 314 "...screen the feasibility from countless potential partner molecules..." This is what I asking about previously in this report.

Line 316 What do they mean by "...thus meaningful for the real applications of molecular cocrystals."?

P325-327 I really didn't see where the morphology control was rationalized to explain anything. There are observations here (and some impressive ones at that), but virtually no explanations.

Notes of interest for the authors:

1. There are 115 entries in the CSD of structures that contain tetracyanobenzene.
2. Anthracene:TCNB has been extensively studied. It has a low temperature form in addition to the room temperature form. Did gthe authors find any evidence of the formation of that form under sublimation conditions (admittedly at much higher temperatures).

Reviewer #3 (Remarks to the Author):

This is a very interesting manuscript that describes a creative method to form cocrystals. The method is based on sublimation and there are some interesting observations in terms of optical properties. It is very true that there is a lack of a method to form cocrystals by way of sublimation (there are a few reports here and there).

Overall, I am generally supportive of the work being published in Nature Communications. The work will be welcomed.

A few suggestions:

- i) Did the authors attempt to form the new cocrystals from solution? It is important to emphasize the point.
- ii) Is it possible to form hydrogen-bonded cocrystals using this method? Many cocrystals (especially pharmaceuticals) are held together by hydrogen bonds. The authors should address the point as a possible limitation and/or area for future work.
- iii) Past work that addresses an intrinsic need to develop methods to grow cocrystals should be cited. Early work of MacGillivray in J. Am. Chem. Soc. 2007, 129, 32 using sonocrystallization is such a work.

Response to Reviewers comments:

We sincerely thank all the reviewers for spending their precious time to review our manuscript and offering us with constructive suggestions to improve the quality of this work. Given below are our point-by-point responses to the comments and suggestions.

Response to the Comments and Suggestions of Reviewer 1

The reviewer's comments: *"The manuscript, '1D Versus 2D Cocrystals Growth via Microspacing In-Air Sublimation' describes a very simple and efficient method to crystallize the same binary organic compounds in two different morphology. As authors correctly stated the physical properties depend not only on the choice of molecules but also on the morphology of thin films. These are interesting experimental results."*

We are very grateful to the reviewer's appreciation of our work, and to the helpful suggestions.

The reviewer raised *few points may be discussed in more details.*

1. **The reviewer commented:** *Authors claim that the crystallization of co-crystals is "still in its primary stage" and cited the ref. [17]. Just during the last few years, some detailed papers appeared in Crystal Growth and Design reporting charge transfer compounds crystallization. Charge transfer compounds are co-crystals, therefore, authors should refer to these papers and including these papers in their discussion. It is also difficult to agree with authors claim that "Unfortunately, little advances have been achieved in this respect, wherein a complex physical vapor transport (PVT) conducted in vacuum or inert gas was commonly adopted". PVT in an inert gas is not a complex technique and the progress in physical property improvements of organic semiconductors made from charge transfer compounds are significant.*

Authors reply: We thank the reviewer for the kindly reminder of the improper description about the current status of cocrystals and of the PVT method in our original manuscript. So, we revised the original description into **"Along with the increasing number of experimental case studies of cocrystal complexes, the demand for more effective growth method of cocrystals is also increasing"**, and **"Significant improvements in physical property of semiconducting charge transfer cocrystals have been made** by adopting physical vapor transport (PVT) technique". Additionally, the following papers (Solvent-dependent stoichiometry in perylene-7, 7, 8, 8-tetracyanoquinodimethane charge transfer compound single crystals, *Cryst. Growth Des.*, 2014, 14 (12), 6376-6382; Molecular Marriage via Charge Transfer Interaction in Organic Charge Transfer Co-Crystals toward Solid-State Fluorescence Modulation, *Cryst. Growth Des.*, 2017, 17 (3), 1251-1257; Impact of C-H \cdots X (X = F, N) and π - π

Interactions on Tuning the Degree of Charge Transfer in F6TNAP-Based Organic Binary Compound Single Crystals, *Cryst. Growth Des.*, **2018**, *18*, 1776–1785; Structure, Stoichiometry, and Charge Transfer in Cocrystals of Perylene with TCNQ-Fx, *Cryst. Growth Des.*, **2016**, *16*, 3028–3036; Crystal Growth, HOMO–LUMO Engineering, and Charge Transfer Degree in Perylene-FxTCNQ (x = 1, 2, 4) Organic Charge Transfer Binary Compounds, *Cryst. Growth Des.*, **2016**, *16*, 3019–3027;) are added to the reference list in the new manuscript as Ref. 20 to 21 and 25 to 27.

2. **The reviewer’s concern:** “*The microspacing in-air sublimation (MAS) is definitely a very simple method, however, due to possible oxidation of some air sensitive molecules it has specific technological limitation only very shortly discussed in this paper.*”

Authors reply: Thanks for the recognition of the method. Because of the well-known instability of organic materials under high temperature, actually the oxidation and degradation of organic molecules during the in-air sublimation growth was also the most concern of us. So, from the beginning of our study about this method, we have conducted plenty of measurements to check the stability of the materials in the growth procedures. Here we take pentacene and rubrene as examples, both of which are the most representative organic semiconductors with high carrier mobility, and are recognized to be air-sensitive. **Figure R1** shows the thermal analysis results of rubrene and pentacene, where we can see their melting points are quite close to the decomposition temperature. (For pentacene we cannot even observe a melting point before severe weight loss.)

Figure R1. Thermal analysis of rubrene (a-d) and pentacene (e-h). (a, c, e and g) Differential scanning calorimetry (DSC)-thermal gravimetric analysis (TGA) curves of rubrene and pentacene powders at a rate of 10 °C min⁻¹. The red line is the TG curve and the black one is DSC curve. (b, d, f and h) DSC curves of rubrene and pentacene at a rate of 10 °C min⁻¹.

To verify the stability of rubrene and pentacene in the process of in-air sublimation, we measured and compared the IR spectra, the MALDI-TOF-MS (matrix assisted laser desorption ionization time of flight mass spectrometry), ¹H NMR spectra, ¹³C

NMR spectra, and elemental analysis of the starting materials (purchased from TCI with sublimed grade purity) and crystals grown by microspacing in-air sublimation. As shown in **Figure R2**, there were no notable difference between the starting materials and the grown crystals. These results proved the identity and purity of the grown crystals, and demonstrated that rubrene and pentacene crystals can be grown by our method without oxidation and degradation.

Figure R2. IR spectra (a), elemental analysis data (b), ¹³C NMR (c), ¹H NMR (d) and MALDI-TOF (e) spectra of the starting materials and grown crystals (obtained by scraping the materials from the upper substrate, containing solidified melt) of rubrene and pentacene. (The solubility of pentacene is too low to do MALDI-TOF and NMR tests)

We have discussed that a short source-substrate distance and fast evaporation, these factors limited the amount of oxygen in the confined space to prevent oxidation. And moreover, the relative lower sublimation temperature is another important attributor. The sublimation temperature is normally about one hundred or more lower than the decomposition temperature of the organic materials. *E.g.*, pentacene starts weight loss at 355 °C in air atmosphere (**Figure R1**); while the growth temperature in our method is at 255 °C, which is much lower than the decomposition temperature. The details about the stability of the organic materials during the sublimation process have been discussed in our last paper. The identity and purity characterizations of the grown cocrystals are shown in **Supplementary Figure 2** of this work. Here according to the reviewer’s reminder of “*only very shortly discussed in this paper*”, we add the following sentences “**More detailed discussions about the stability of the organic materials during the sublimation process refer to our last work.²⁹**” to the main text, Line 118, Page 4. Anyhow the concern of the reviewer is reasonable. Actually, we also have planned in the future work that for the molecules which are extremely

sensitive to air, we could conduct the growth under inert atmosphere (e.g. in the glove box) to avoid oxidation.

3. **The reviewer asked** “*In the experiment description, authors write the bottom and top substrate was separated by “a tiny space of 150 μm ”. What was the average grain dimensions of used powder in both experiments when a grinded or a mixture of two powder is used? How the powder was “simultaneously located” on the top substrate? Figure 1 is confusing. The section 1a suggest that these two compounds are widely separated from each other. Therefore, one or both components need to travel long horizontal distances to form a homogeneous co-crystal with constant stoichiometry. Is it the case? Authors’ commentary on this would be helpful.*”

Authors reply: As shown in **Figure R3a**, for the case of a mixture of two powders, the starting materials of the two compounds are microcrystals. The size of TCNB powder is in the range of several tens of micrometers, and the size of needle-like fluoranthene powder is about several hundreds of micrometers in length and tens of micrometers in diameter. In our experiment the top and bottom substrates are separated by two glass sleepers with thickness of 150 μm . We can press the top substrate after placing it on the separating sleepers to ensure an even space distance between them. It is allowed to let the top substrate to directly contact with some fluoranthene powders in the beginning of the sublimation growth, because fluoranthene deposited on the top substrate will melt into liquid in the later process. In that process the melt no longer contacts with the bottom substrate. For the case of grinded mixtures, the grain size is much smaller, with average grain dimensions of several micrometers (**Figure R3b**). The grain dimensions of the starting materials can be also witnessed from **Figure 3h** and **k** in the manuscript.

Figure R3. Microscope images of the mixed (a) and the grinded mixed (b) starting materials.

“Simultaneously located” was meant to express that the two kinds of powders were located closely on the bottom substrate. The related schematic of Figure 1a thus leads misunderstanding that these two compounds were widely separated from each other. In actual experiments the two kinds of powders were just put together on the substrate, but without grinding. So, there is no need for the two components to travel long

horizontal distances to form cocrystals. Accordingly, we revised the original Figure 1a into the one as in **Figure R4**; and revised the expression “simultaneously located” into “located together”.

Figure R4. MAS apparatus for growth of 1D FTCs.

4. **The reviewer commented** “Authors claim that a convection flow should be observed according to calculated Rayleigh number. However, the values of these number are not given. Also, the value of Rayleigh number limiting laminar flow from convection could be useful for process understanding.”

Authors reply: Thanks for the suggestion. According to the formula: $Ra = \frac{g\beta\Delta T h^3}{\nu\kappa}$, and Ref. [33], Kloc, C.; Simpkins, P.; Siegrist, T.; Laudise, R., *Journal of crystal growth* **1997**, 182, 416-427], a microspacing distance of 150 μm leads to a Rayleigh number estimated to be 1.7×10^{-7} . According to Ref. [33] we revised “convection” into “molecular flow vapor transport”. So, the original description is revised into “a microspacing distance of 150 μm leads to a small Rayleigh number estimated to be 1.7×10^{-7} ”, and “An efficient buoyancy driven molecular flow vapor transport mode would be generated within the confined space between two substrates”.

5. **The reviewer commented** “In line 121, authors write that crystallization temperature was 140 °C. However, in line 93 the bottom substrate temperature was 130 °C. It seems that it is hard to decide if the morphology of powders or the substrate temperature was responsible for needle or platelet growth. The conclusion of this paper could be found at the end of the paper and it is: ‘In situ observation of the crystallization processes revealed two-step sublimation for 1D growth while synchronous sublimation for 2D growth’ Experimental results from these papers seems to confirm this conclusion. Especially, detailed described growth mechanism by sequential evaporation of individual components for 1D morphology and simultaneous evaporation of components for 2D morphology is the leading thought constructing this paper.”

Authors reply: Line 93, “The growth procedure was implemented by heating the bottom substrate at about 130 °C for 30 min.” Here the “130 °C” refers to a condition in general. In actual growth of 2D cocrystals as we described in Line 127, the formed charge transfer complex has a relative higher sublimation temperature of 140 °C. So, to avoid misunderstanding, we revise the description in Line 98 into “by heating the bottom substrate at 130 or 140 °C”. The reviewer’s conclusion about the growth mechanism by “sequential evaporation” and “simultaneous evaporation” is really the key points we want to present. Thanks.

Response to the Comments and Suggestions of Reviewer 2

The reviewer's comments: *"I am reluctant to use the relatively noncommittal adjective 'intriguing' to describe this manuscript. However, its use is definitely meant in the positive sense. As a long time player in the discipline of organic crystals, and with some experience in the kind of charge transfer complexes described here I found the results presented both surprising, and yet frankly questioned why this effect had not been previously observed."*

We are deeply grateful to the reviewer's high appreciation of our work. It is really our great honor to be acknowledged and accepted by an expert in the discipline of organic crystals.

1. **The reviewer commented:** The authors present rather convincing experimental evidence to support their claims. While I had doubts about the general application of this sublimation phenomenon to other systems, the two final examples of pyrene-TCNB and anthracene-TCNB did serve to allay some of those doubts. Nevertheless, I found myself asking the question of whether any other acceptor would behave the same way. Some other questions I found myself asking: Does there have to be some compatibility between the vapor pressures of the two components? What would be the result of an experiment in which, say, the two different preparations of the two components were arranged in opposite corners of a square arrangement? Would a sample of mixed crystal habits be obtained? How strictly must the experimental parameters be maintained in order to maintain control over the product? As noted above: How general is this phenomenon? Is it limited to pi charge transfer complexes? If so, do the authors have any insight into which donors and which acceptors? Would hydrogen bonding co-crystals be suitable candidates for such experiments? Are there some lower limits on the vapor pressure of one or both of the solid components?

As noted above, most of the questions here fall into the "intriguing" category, rather than criticism; nevertheless, I felt that I was looking for answers. On the other hand, the results presented here certainly suggest new areas to explore and the paper would serve as a catalyst for those studies. May TCNB is not the only pi-acceptor that would exhibit this controlled habit change under different sublimation conditions. In that sense, the paper is worthy of publication – as a catalyst for further work in an area that has not yet been explored.

Authors reply: Thanks again for giving us so high marks.

- (1) Since the reviewer is mostly interested about the general application of our sublimation growth method to other systems, e.g., the hydrogen-bonded cocrystals. And Reviewer 3 also raised the same interest: *"Is it possible to form hydrogen-bonded cocrystals using this method? Many cocrystals (especially pharmaceuticals) are held together by hydrogen bonds. The authors should address the point as a possible*

limitation and/or area for future work.” Thus, we conducted a series of supplementary experiments to check the universality of the method.

We choose carbamazepine (CBZ)-based cocrystals to serve as a model system of hydrogen-bonded cocrystals. Carbamazepine is a widely used antiepileptic drug whose low aqueous solubility requires to be improved for therapeutic efficacy. Cocrystals of carbamazepine typically form through hydrogen bond of the primary amide group with a conformer. (**Figure R5a and b**) Pharmacokinetic studies of carbamazepine-saccharin (CBZ-SAC) cocrystal have shown blood level increases due to dissolution improvement over the marketed pure drug. (Hickey, M.B. et al., Performance comparison of a cocrystal of carbamazepine with marketed product. *Eur. J. Pharma. Biopharma.*, **2007**, *67*, 112–119). Carbamazepine–saccharin cocrystal was prepared conventionally by cooling crystallization from alcohol solution. Here we adopt our microspacing in-air sublimation to grow it. As shown in **Figure R5c**, our method works well for CBZ–SAC cocrystal, obtaining plate-shaped single crystals with size of about 10-30 μm . The XRD pattern of the grown cocrystal is indexed to a triclinic $P\bar{1}$ polymorph. And moreover, it indicates an oriented growth on the substrate, with c-axis perpendicular to the substrate surface. Besides CBZ–SAC, we have also tried to grow another carbamazepine-nicotinamide (CBZ–NCT) cocrystal. As shown in the right column of **Figure R5**, needle-like CBZ–NCT cocrystal formed on the top substrate, similarly with oriented growth habit. Maybe because of the strong intermolecular hydrogen bonds of this kind of cocrystals, morphology control over the cocrystal is not easy to be achieved. Anyhow these results prove the general application of the method in growth of hydrogen-bonded pharmaceutical cocrystals. Researchers could use the method to screen cocrystal formation of drugs without inclusion of solvent molecules.

Figure R5. Chemical structures (a), single crystal structures (b), microscope images (c) and XRD patterns (d) of CBZ–SAC (left) and CBZ–NCT (right) cocrystals grown by MAS.

In addition to TCNB-based complex and hydrogen-bonded pharmaceutical cocrystals, we have also used our MAS method to grow cocrystals formed between polycyclic aromatic hydrocarbons (PAH) and haloperfluoroarenes. As shown in **Figure R6**, pyrene–1,3,5-trifluoro-2,4,6-triiodobenzene (pyrene–TIFB) cocrystals have been successfully grown on the substrate, with morphology control from 1D to 2D by changing the growth conditions. Growth at relative lower temperature (120 °C) produced 1D needle-like cocrystals (**Figure R6a**). On the other hand, thoroughly mixing of the two constituent molecules through grinding and growth at relative higher temperature (140 °C) is more conducive to obtain 2D plate-like cocrystals (**Figure R6b**). The XRD patterns (**Figure R6e**) and Raman spectra (**Figure R6f**) of the as-grown pyrene–TIFB cocrystals are distinct from that of each constituent molecule, suggesting the formation of cocrystals. The difference of relative peak intensity in Raman spectra of the needle-like and plate-like cocrystals may be related with different morphology.

Figure R6. Microscope images (a and b) and fluorescence microscope images (b and d), XRD pattern (e) and Raman spectra (f) of 1D needle-like and 2D plate-like pyrene-TIFB cocrystals. (Scale bar: 50 μm for a and c, 20 μm for others)

Figure R7. Microscope images (a, c) and fluorescence microscope images (b, d), XRD pattern (e) and Raman spectra (f) of 1D needle-like anthracene-OFN cocrystals (a-b) and 2D plate-like (c-d) anthracene crystals. (Scale bar: 100 μm for a and b, 50 μm for others)

The growth of anthracene-octafluoronaphthalene (anthracene-OFN) cocrystals exhibited another notable phenomenon. When growing at relative lower temperature (100 $^{\circ}\text{C}$), 1D needles crystalized on the substrate (**Figure R7a**); when growing at relative higher temperature (higher than 120 $^{\circ}\text{C}$), 2D plates formed (**Figure R7c**). However, it is not the case that morphology control is realized on anthracene-OFN. According to the XRD pattern and Raman spectra in **Figure R7e-f**, the 1D needles are indeed cocrystals, with oriented growth of b-axis perpendicular to the substrate surface. But the Raman spectra of 2D plates shows that they actually are anthracene

crystals. We then checked the in situ observed formation process of 2D plated crystals, which reveals that during the heating process the needle-like cocrystals decomposed and then plated anthracene crystals appeared. (**Figure R8**) We believe this phenomenon is corresponding to what the Reviewer has proposed “Does there have to be some compatibility between the vapor pressures of the two components? Are there some lower limits on the vapor pressure of one or both of the solid components?” The transformation from anthracene–OFN cocrystal to anthracene crystal is attributed to the incompatibility between the vapor pressures of the two components at high temperature. OFN, as a derivative of naphthalene, has low melting point (87-88 °C) and high vapor pressure, which makes it being easy to sublime even at room temperature. On the other hand, anthracene has relatively high melting point (215 °C) and low vapor pressure. Thus, with the increasing of temperature OFN will volatilize from the cocrystal and the left anthracene recrystallizes on the substrate.

Figure R8 Microscope images of in situ observation of transformation from needle-like cocrystals to plate-like anthracene crystals during the heating process. (Scale bar: 100 μm)

The results demonstrate that our MAS method has a broad universality in cocrystals growth that even constituent molecules with great different vapor pressure work well. But, as reminded by the Reviewer, the compatibility between the vapor pressures of the two components indeed plays an important role in the applicability of morphology control by MAS.

Thus, we change the subheading on Line 280 to “**Generality of cocrystals growth by MAS.**” and add the following sentences on Line 293 “**Besides the PAH–TCNB complexes, we also tested other cocrystals to check the generality of the method, including PAH–haloperfluoroarene systems and even hydrogen-bonded pharmaceutical cocrystals. A typical carbamazepine-saccharin⁴⁹ pharmaceutical cocrystal, which was conventionally prepared from solution, was employed in the MAS growth. As shown in Fig. 5b, plate-shaped carbamazepine-saccharin cocrystal with size of about 10-30 μm was grown on the substrate. Its XRD pattern is indexed to a triclinic $P\bar{1}$ polymorph, and indicates an oriented growth with c-axis perpendicular to the substrate surface (Fig. 5c). Another pharmaceutical cocrystal carbamazepine-nicotinamide⁵⁰ is also applicable to the method, for which needle-like cocrystal with oriented growth was obtained (Supplementary Figure 10). For the PAH–haloperfluoroarene systems, pyrene–1,3,5-trifluoro-2,4,6-triiodobenzene⁵¹ cocrystal was successfully grown on the substrate, with morphology control from 1D to 2D by changing the growth conditions likewise (Fig. 5d–h and Supplementary**

Figure 11). Whereas for the anthracene–octafluoronaphthalene⁵² complex, growth at relative lower temperature did produce 1D needle-like cocrystal; growth at relative higher temperature (higher than 120 °C) obtained 2D plated anthracene crystal, but not cocrystal (Supplementary Figure 12). The failure of morphology control over anthracene–octafluoronaphthalene cocrystal is attributed to the incompatibility between the vapor pressures of the two components, where the low melting point (87-88 °C) and high vapor pressure of octafluoronaphthalene make it volatilize from the cocrystal with temperature increasing (Supplementary Figure 13). The results demonstrate that our MAS method has a broad universality in cocrystals growth that even hydrogen-bonded pharmaceutical cocrystals are applicable. Morphology control at high temperature should be aware of the compatibility between the vapor pressures of the constituent molecules.

In the conclusion part, Line 357, the following sentence is added “The method has a broad universality in cocrystals growth that even hydrogen-bonded pharmaceutical cocrystals are applicable.

Figure R9 is added to the main text as **Fig. 5**.

Figure R5-8 are added to the Supplementary Information as Supplementary Figure 10-13.

We add the following sentences in the “Growth of single cocrystals” of the Methods part. “carbamazepine-saccharin (160 °C), carbamazepine-nicotinamide (140 °C), pyrene-TIFB (120 °C for needle shaped and 140 °C for plate shape, with the space distance of 1 mm), anthracene-OFN (100 °C).”

Figure R9. Generality of MAS in growth of hydrogen-bonded pharmaceutical cocrystals and PAH–haloperfluoroarene complex. Chemical structures of carbamazepine-saccharin (a) and pyrene-TIFB (d). Microscope image (b) and XRD patterns (c) of the as-grown carbamazepine-saccharin cocrystals. Microscope images

(e and f) and fluorescence microscope images (g and h) of 1D needle-like and 2D plate-like pyrene-TIFB cocrystals.

- (2) The reviewer asked questions about the “*experimental parameters*”, including “*How strictly must the experimental parameters be maintained in order to maintain control over the product? How critical is the space h?*”. He/she suggested that “*the history and the success rate need to be included here*”, because “*Anybody involved in growing crystals as part of their experimental activities knows that not every crystallization works, even ones that have been done many times.*”

Thanks for the comments and the kindly reminder. Firstly, in general, growth of cocrystals by MAS is quite easy. If irrespective of crystallinity and morphology of the as-grown cocrystals, the success rate is nearly 100% in our experiments.

Secondly, the experimental parameters, including the space h , growth temperature and the mixing and distribution state of starting materials, indeed have great effect on the crystallinity and morphology of the cocrystals. For the space h , as pointed by the reviewer, we found it not only influences the vapor transport condition, but also determines the temperature of the top substrate by heat transfer through the micro space. As shown in **Figure R10**, we have tested the temperature dependence of the top substrate on that of the bottom substrate with different spacing distances (h) during the temperature rising and holding process. We can see when h is below 400 μm the temperature of the top substrate raises simultaneously with that of the bottom substrate. At a 150 μm of h , the top substrate could be stabilized at 270 $^{\circ}\text{C}$ when the bottom substrate was heated to 280 $^{\circ}\text{C}$, with a temperature difference of only 10 $^{\circ}\text{C}$. While when h was set at 4 mm, the temperature of the top substrate lagged obviously behind that of the bottom substrate, with a temperature difference of as much as 140 $^{\circ}\text{C}$.

Figure R10 Temperature curve of top substrate of different spacing distance.

As discussed in the manuscript, the temperature of the top substrate has essential

effect on the cocrystal morphology, and actually also on the crystallinity. Figure R11 shows the growth of 1D fluoranthene–TCNB cocrystal as an example. When the space h is too small (In this case the top substrate is prone to directly contact with the melt, so we label it as $0\ \mu\text{m}$), cocrystal is hard to crystalize from the melt, resulting many glassy state solidified residuals (**Figure R11a**). When the space h is of about $400\ \mu\text{m}$, the crystallinity is okay, but maybe because of too many of nucleuses the length of the grown cocrystals is much shorter than that in a case of $150\ \mu\text{m}$ (**Figure R11b**). For the case of an even larger h of $1\ \text{mm}$, the grown cocrystals are no more isolated single crystals but dendritic polycrystals (**Figure R11c**).

Figure R11 Fluorescence microscope images of cocrystals grown by MAS with different space h . (a) For a small space h , the top substrate is prone to directly contact with the melt. (So we label it as $0\ \mu\text{m}$.) Cocrystal is hard to crystalize from the melt, resulting many glassy state solidified residuals. (b) For a space h is of $\sim 400\ \mu\text{m}$, the crystallinity is okay, but maybe because of too many of nucleuses the length of the grown cocrystals is much shorter than that in a case of $150\ \mu\text{m}$. (c) For the case of an larger h of $1\ \text{mm}$, the grown cocrystals are no more isolated single crystals but dendritic polycrystals.

As workers on organic crystals, we deeply understand the reviewer’s comments that “*Anybody involved in growing crystals as part of their experimental activities knows that not every crystallization works, even ones that have been done many times.*”. In our experiments, growth of pure 1D needle-like cocrystals is relatively easy to realize (**Figure R12a**); while growth of 2D plate-like cocrystals is frequently contaminated with short rod-shaped cocrystals formed on the edge of the main growth area. (**Figure R12b**)

Figure R12 Fluorescence microscope images of 1D and 2D cocrystals in large-scale view.

Whereas a space of 150 μm is not always the most optimal choice for all materials. E.g., for pyrene-TIFB cocrystal which has poor crystallization capacity even from the solution, we found that a relative larger h (~ 1 mm) produced higher quality cocrystals. Thus, we add the following sentences in the “Growth of single cocrystals” of the Methods part. “Selection of optimal experimental parameters for different materials, including the space h , growth temperature and the mixing and distribution state of starting materials, has essential effect on the cocrystal morphology and crystallinity. E.g. an improper space h would result poor quality of the grown cocrystals (Supplementary Figure 14)” **Figure R11** is added as Supplementary Figure 14 in the Supplementary Information.

- (3) The review gave us Notes of interest that “There are 115 entries in the CSD of structures that contain tetracyanobenzene. Anthracene:TCNB has a low temperature form in addition to the room temperature form. Did the authors find any evidence of the formation of that form under sublimation conditions (admittedly at much higher temperatures).” We thank the reviewer’s kindly notes. We have queried the data base and compared our structure with the reported ones. The low and high temperature forms of anthracene:TCNB belong to different space groups, $P2_1/a$ for the low temperature phase and Cm for the high temperature one. However their unit cell parameters are quite similar: for the low temperature form, $a = 9.441(2)$ Å, $b = 12.650(4)$ Å, $c = 7.299(1)$ Å, and $\beta = 93.11(2)^\circ$; for the high temperature form, $a = 9.505(1)$ Å, $b = 12.748(2)$ Å, $c = 7.417(2)$ Å, and $\beta = 92.45(2)^\circ$. So their XRD patterns are also very similar to each other. According to the comparison of XRD profiles shown in **Figure R13**, the anthracene:TCNB cocrystal grown by MAS should be the high temperature form based on the diffraction peak corresponding to (002) diffraction. In our future work we will also explore the feasibility of polymorph control by MAS.

Figure R13 XRD patterns of Anthracene:TCNB grown by MAS, and the calculated pattern for the two form.

2. Revisions on writings:

- 1) Line 14: the word “new and” is deleted.
- 2) Line 18 and Line 25: “which is essential for structure property relations” is added; “growth kinetics-defined crystal habit” is added in Line 25.
- 3) Line 25: structural inherence” is changed into “growth kinetics-defined crystal habit”.
- 4) Line 30: “as” is deleted.
- 5) Line 38: “providing” is changed into “provide”.
- 6) Line 42: Three review papers by Zaworotko and Almarrson are added as Ref. [7] to Ref. [9] (Synthesis and structural characterization of cocrystals and pharmaceutical cocrystals: mechanochemistry vs slow evaporation from solution. *Crystal Growth and Design* 2009, 9 (2), 1106-1123; The role of cocrystals in pharmaceutical science. *Drug discovery today* 2008, 13 (9-10), 440-446; Pharmaceutical cocrystals: along the path to improved medicines. *Chemical communications* 2016, 52 (4), 640-655.).
- 7) Line 45: “applications in many other fields” is changed into “in many fields”.
- 8) Line 47: The original sentence is changed into: “The co-assembly of multicomponents has been thoroughly investigated in the context of supramolecular chemistry and molecular crystal engineering”.
- 9) Line 58: “resulted” is deleted.
- 10) Line 63: “When considering the ease of integration into devices...” means crystals with proper size and that are grown directly on the substrates such as silicon wafers and ITO glasses are more convenient to be integrated with electrodes and other components in the devices.
- 11) Line 61: The sentence is revised into “Significant improvements in physical property of semiconducting charge transfer cocrystals have been made”. Here the revision is also referred to Reviewer 1.
- 12) Line 66: “molecular packing symmetry” is changed into “crystal habit”; “hardly affected through man-made regulations” is changed to “is hardly to be altered”.
- 13) Line 71: the sentence is changed into: “Because of the microspacing distance between the source and growth position in the novel growth technique of MAS, we realize morphology control over the cocrystals of different CT complexes.”.
- 14) Line 74: “(with grinding mix or not)” is changed into “(mixing by grinding or blend without grinding)”.
- 15) Line 79: “mechanism” is changed to “conditions”.
- 16) Line 84: To illustrate why we choose the TCNB-based systems, the following sentences are added: “TCNB-based cocrystals generally exhibit enhanced luminescence owing to the CT transition from π -conjugated electron donor to the TCNB electron acceptor, which have been studied as models of light-emitting and harvesting systems.^{21,30-32} Moreover, the distinct luminescence color for TCNB-based cocrystals with respect to one-component crystals facilitates recognition of the formation of cocrystals. So considering that fluoranthene crystal possesses obvious blue fluorescence”.
- 17) Line 95: “simultaneously located” is changed into “located together”.
- 18) Line 99: “down” is changed into “lower”.

- 19) Line 109: The application of the equation under the conditions of this system is mainly referred to the pioneer works of R.A. Laudise and Ch. Kloc, et.al. (Kloc, C.; Simpkins, P.; Siegrist, T.; Laudise, R., Physical vapor growth of centimeter-sized crystals of α -hexathiophene. Journal of crystal growth 1997, 182, 416-427. and Laudise R.; Kloc Ch.; Simpkins P.; Siegrist T.; Physical vapor growth of organic semiconductors. Journal of crystal growth 1998, 187, 449-454.) They used such equation to model transport and growth. The buoyancy-driven flow in this system can be compared with that in their systems.
- 20) Line 121: The original sentence is changed into: "The identity and purity characterizations of the grown FTCs are shown in Supplementary Figure 2."
- 21) Line 123: The instructions are revised into: "a key factor is the way how the two component molecules are mixed and distributed on the bottom substrate. That is, when the two kinds of component molecules were just put closely together without being ground, the cocrystals grown at 130 °C show 1D needle-like shapes on the top substrate."
- 22) Line 128: "peculiar" is deleted. (It was meant to say that a 2D morphology is unfrequent for the PAH-TCNB cocrystals grown by other techniques.)
- 23) Line 130: "As we can see" is changed into "As shown in".
- 24) Line 138: The spelling of "fluoranthene" is corrected.
- 25) Line 145: The word "small" is deleted.
- 26) Line 154: The face is re-labeled as "(001) face". We agreed with the Reviewer that although the 001 reflection is absent by symmetry in XRD, the face is still there;
- 27) Line 165: "the individual components" is changed into "the two individual constituent molecules"; both the fluorescence data of the cocrystal and the two individual constituent molecules are measured by ourselves, and are shown in Supplementary Information Fig. S5a. The quantum yield (Φ_f) of fluoranthene is according to the paper (Elliott, E. L.; Orita, A.; Hasegawa, D.; Gantzel, P.; Otera, J.; Siegel, J. S. J. O.; chemistry, b., Synthesis and properties of 1, 6, 7, 12, 13, 18, 19, 24-octamethylacenaphthylene [b, l] tetraphenylene. 2005, 3 (4), 581-583.). So we add the paper as Ref. [38] in the new version of the manuscript.
- 28) Line 256: The Reviewer asked "Why is the crystallization from the liquid melt more likely to be a spontaneous process?" We want to say that because the melting point of the formed cocrystals is higher than the temperature of the liquid melt, probably the crystallization occurred spontaneously, just like the crystallization from supersaturated solutions.
- 29) Line 257: "inherited from the intrinsic crystal structure" is changed into "to be defined by the intrinsic structural symmetry".
- 30) Line 272: "of each crystal facets" is changed into "of each of the crystal faces".
- 31) Line 289 "I" after Figure 4 is deleted.
- 32) Line 341: About "active electric disturbance" and "optical scattering modules". In some optoelectronic devices, e.g., OLEDs and OPVs, scattering particles or surfaces are used to improve light output and to enhance optical absorption, respectively. Patterned holes on one side of the cocrystals may be used as such kind of scattering modules in future single crystal devices. And because the

patterns have specular positions and directions, the modulation of light may be more precise. To be prudent, “active electric disturbance” is deleted.

33) Line 343: “much needed” is changed into “required”.

34) Line 354: “...thus meaningful for the real applications of molecular cocrystals” is meant to say that the new method to grow cocrystals on substrate and to modulate their morphology should be helpful for the applications of cocrystals in optical or electronic devices.

35) Line 365: “where the morphology control is rationalized to be resulted from the competition effect between structure inherence and the thermodynamics driving force by aid of energy calculations” is deleted.

We are so grateful for these detailed comments. They not only improve this paper, but also help us a lot for our future scientific career.

Response to the Comments and Suggestions of Reviewer 3

The reviewer’s comments: *“This is a very interesting manuscript that describes a creative method to form cocrystals. The method is based on sublimation and there are some interesting observations in terms of optical properties. It is very true that there is a lack of a method to form cocrystals by way of sublimation (there are a few reports here and there). Overall, I am generally supportive of the work being published in Nature Communications. The work will be welcomed.”*

We are deeply grateful to the reviewer’s high appreciation of our work. Thanks for the high remarks and recommendation of publication in Nature Communications.

Response to the suggestions of the Reviewer:

1. **The reviewer suggested** “Did the authors attempt to form the new cocrystals from solution? It is important to emphasize the point.”

Figure R14. Fluorescence microscope images of cocrystals of (a) fluoranthene-TCNB, (b) pyrene-TCNB^[1] and (c) anthracene-TCNB^[2] grown from solution.

Thanks for the suggestion. Yes, we have tried to grow the new fluoranthene-TCNB

cocrystal from dichloromethane solution of mixed constituent molecules. As shown in **Figure R14a**, growth of cocrystals from solution only produced 1D needle-like morphology, which is consistent with the previous results reported by other groups that pyrene-TCNB and anthracene-TCNB cocrystals grown from acetonitrile solutions showed similar morphology of 1D needle-like. In the main text of the manuscript we have involved related discussions. Thus according to the Reviewer's suggestion, after "thus resulting crystal habit to be defined by the attachment energy between lattice layers just like those grown from solutions."^{42, 46-48}, we add a sentence: (FTCs grown from solution show 1D morphology. Supplementary Figure 9). **Figure R14a** is added into Supplementary Information as Supplementary Figure 9.

[1] Sun Y, Lei Y, Liao L, et al. Competition between Arene-Perfluoroarene and Charge-Transfer Interactions in Organic Light-Harvesting Systems[J]. *Angewandte Chemie International Edition*, 2017, 56(35): 10352-10356.

[2] Lei Y L, Liao L S, Lee S T. Selective growth of dual-color-emitting heterogeneous microdumbbells composed of organic charge-transfer complexes[J]. *Journal of the American Chemical Society*, 2013, 135(10): 3744-3747.

Here the Ref. [1] and Ref. [2] are Ref. [46] and Ref. [42] in the manuscript.

2. **The reviewer suggested that** "Is it possible to form hydrogen-bonded cocrystals using this method? Many cocrystals (especially pharmaceuticals) are held together by hydrogen bonds. The authors should address the point as a possible limitation and/or area for future work."

Thanks for the suggestion. The concern about the general application of our method to other systems is also raised by Reviewer 2. Here, we choose carbamazepine (CBZ)-based cocrystals as a model system of hydrogen-bonded cocrystals to check the universality of the method. Carbamazepine is a widely used antiepileptic drug whose low aqueous solubility requires to be improved for therapeutic efficacy. Cocrystals of carbamazepine typically form through hydrogen bond of the primary amide group with a conformer (**Figure R15a** and **b**), and were prepared conventionally by cooling crystallization from solutions. Hereinto carbamazepine-saccharin (CBZ-SAC) cocrystal is a representative one whose pharmacokinetic performances have been studied and compared over the marketed pure drug. Here we adopt our microspacing in-air sublimation to grow it. As shown in **Figure R15c**, our method works well for CBZ-SAC cocrystal, obtaining plate-shaped single crystals with size of about 10–30 μm . The XRD pattern of the grown cocrystal is indexed to a triclinic $P\bar{1}$ polymorph. And moreover, it indicates an oriented growth on the substrate, with c -axis perpendicular to the substrate surface. Besides CBZ-SAC, we have also tried to grow another carbamazepine-nicotinamide (CBZ-NCT) cocrystal. As shown in the right column of **Figure R15**, needle-like CBZ-NCT cocrystal formed on the top substrate, similarly with oriented growth habit. Maybe because of the strong intermolecular hydrogen bonds of hydrogen-bonded cocrystals,

morphology control over the cocrystal is not easy to be achieved. Anyhow these results prove the general application of the method in growth of hydrogen-bonded pharmaceutical cocrystals. Researchers could use the method to screen cocrystal formation of drugs without inclusion of solvent molecules.

In addition to TCNB-based complex and hydrogen-bonded pharmaceutical cocrystals, we have also used our MAS method to grow cocrystals formed between polycyclic aromatic hydrocarbons (PAH) and haloperfluoroarenes. Pyrene-1,3,5-trifluoro-2,4,6-triiodobenzene (pyrene-TIFB) (**Figure R16d-h**) and anthracene-octafluoronaphthalene (anthracene-OFN) (**Figure R7**) cocrystals have been successfully grown on the substrate by MAS. As shown in the right column of **Figure R16**, the morphology of pyrene-TIFB cocrystals can be controlled from 1D needles to 2D plates by changing the growth conditions. The results demonstrate that our MAS method has a nice universality in cocrystals growth.

Figure R15. Chemical structures (a), single crystal structures (b), microscope images (c) and XRD patterns (d) of CBZ-SAC (left) and CBZ-NCT (right) cocrystals grown by MAS

Figure R16. Generality of MAS in growth of hydrogen-bonded pharmaceutical cocrystals and PAH–haloperfluoroarene complex. Chemical structures of carbamazepine-saccharin (a) and pyrene-TIFB (d). Microscope image (b) and XRD patterns (c) of the as-grown carbamazepine-saccharin cocrystals. Microscope images (e and f) and fluorescence microscope images (g and h) of 1D needle-like and 2D plate-like pyrene-TIFB cocrystals.

Accordingly, we change the subheading on Line 280 to “**Generality of cocrystals growth by MAS.**” Following Descriptions are add on Line 293: “**Besides the PAH–TCNB complexes, we also tested other cocrystals to check the generality of the method, including PAH–haloperfluoroarene systems and even hydrogen-bonded pharmaceutical cocrystals. A typical carbamazepine-saccharin⁴⁹ pharmaceutical cocrystal, which was conventionally prepared from solution, was employed in the MAS growth. As shown in Fig. 5b, plate-shaped carbamazepine-saccharin cocrystal with size of about 10-30 μm was grown on the substrate. Its XRD pattern is indexed to a triclinic $P\bar{1}$ polymorph, and indicates an oriented growth with c-axis perpendicular to the substrate surface (Fig. 5c). Another pharmaceutical cocrystal carbamazepine-nicotinamide⁵⁰ is also applicable to the method, for which needle-like cocrystal with oriented growth was obtained (Supplementary Figure 10). For the PAH–haloperfluoroarene systems, pyrene–1,3,5-trifluoro-2,4,6-triiodobenzene⁵¹ cocrystal was successfully grown on the substrate, with morphology control from 1D to 2D by changing the growth conditions likewise (Fig. 5d–h and Supplementary Figure 11)... The results demonstrate that our MAS method has a broad universality in cocrystals growth that even hydrogen-bonded pharmaceutical cocrystals are applicable.**

In the conclusion part, Line 357, the following sentence is added “**The method has a broad universality in cocrystals growth that even hydrogen-bonded pharmaceutical cocrystals are applicable.**”

Figure R16 is added to the main text as Fig. 5.

3. **The reviewer suggested that:** “Past work that addresses an intrinsic need to develop methods to grow cocrystals should be cited. Early work of MacGillivray in *J. Am. Chem. Soc.* 2007, 129, 32 using sonocrystallization is such a work.”

Thanks for the suggestion. Following the suggestion of the reviewer, we have added the paper (Preparation and Reactivity of Nanocrystalline Cocrystals Formed via Sonocrystallization, *Journal of the American Chemical Society* 2007, 129(1): 32-33.) as the Ref. [19] in the new version of the manuscript.

REVIEWERS' COMMENTS:

Reviewer #1 (Remarks to the Author):

This is the review of the revised manuscript. Authors discussed all points that reviewer listed in his review. They introduced additional information according to the reviewer suggestions. In some points, they discussed or explained the reason why they used a special approach in the original manuscript. In the current form, the paper is ready for publishing. It doesn't need any additional corrections.

Reviewer #2 (Remarks to the Author):

I have reviewed the responses of the authors to my original review, as well as the revised manuscript.

Regarding the responses to my review (as well as those by the other referees) the authors have made an impressive effort to respond to points I raised, both with clarifications and additional experiments when necessary. My impression is that the manuscript is significantly improved over the previous (albeit commendable) version, and now is definitely suitable for publication in Nature Communications.

I believe that it will generate a considerable amount of new experimentation in a manner similar to that generated by the initial experiments for the preparation of co-crystals by grinding and solvent-assisted grinding.

Reviewer #3 (Remarks to the Author):

I note that the authors have now shown that the method is applicable to hydrogen-bonded cocrystals. This is a welcomed development and strengthens the paper.

I recommend publication.